# Cross-cultural adaptation and validation of the Chinese version of the short-form of the Central Sensitization Inventory (CSI-9) in patients with chronic pain: A single-center study

Dongfeng Liang[ID][1☯]*, Xiangli Yu[2☯], Xiaojie Guo[3☯], Jie Zhang[1], Ronghuan Jiang[3]*

1 Department of Rheumatology and Immunology, the First Medical Center, Chinese PLA General Hospital, Beijing, China, 2 Outpatient Department, the First Medical Center, Chinese PLA General Hospital, Beijing, China, 3 Department of Psychology, the First Medical Center, Chinese PLA General Hospital, Beijing, China

☯ These authors contributed equally to this work.
* liangdongfeng301@foxmail.com (DL); jiangrh55@126.com (RJ)

**Data Availability Statement:** All relevant data are within the paper and its Supporting Information files.

## Abstract

Chronic pain affects more than 30% of the general population. The 9-item Central Sensitization Inventory (CSI-9) is a shortened version of the CSI-25, which is a patient-reported instrument used to screen people at risk of central sensitization (CS). The aim of this study was to cross-culturally adapt and validate a Chinese version of the CSI-9. The Chinese CSI-9 was generated by translation of the original English version, back-translation, cultural adaptation, and revision using the Delphi method. The Chinese CSI-9 was administered to 235 patients with chronic pain and 55 healthy controls. Structural validity (confirmatory factor analysis), construct validity (correlations with other scales), test-retest reliability (intraclass correlation coefficient, ICC), and internal consistency (Cronbach's α) were evaluated. Confirmatory factor analysis was performed using one factor. The Chinese CSI-9 score was positively correlated with the Pain Catastrophic Scale (PCS) total score ($r = 0.463$), PCS subscale scores ($r = 0.347–0.463$), Brief Pain Inventory (BPI) mean item score ($r = 0.524$), BPI total score ($r = 0.773$), and the number of painful sites ($r = 0.451$). The Chinese CSI-9 had excellent test-retest reliability (ICC = 0.958) and excellent internal consistency (Cronbach's α = 0.902 in the overall sample and 0.828 in the chronic pain population). The optimal cut-off value for the Chinese CSI-9 was 18 points. The Chinese CSI-9 had excellent test-retest reliability and satisfactory structural validity and construct validity. The CSI-9 could potentially be utilized in China as a self-report questionnaire in both clinical practice and research settings.

## Introduction

Up to 30% of people are affected by chronic pain [1, 2], which can be caused by musculoskeletal disorders (e.g., low back pain or neck pain and osteoarthritis), fibromyalgia, headache, temporomandibular joint disorder (TMD), and irritable bowel syndrome (IBS) [3]. Chronic

**Funding:** The author(s) received no specific funding for this work.

**Competing interests:** The authors have declared that no competing interests exist.

pain negatively affects the quality of life and is associated with reduced mobility, psychological distress, sleep disturbance, and an economic burden [4]. In China, since chronic pain does not threaten the life of the patients, it is often overlooked, leading to low diagnosis rates, low treatment rates, insufficient pain control, poor management of pain, and neglect of pain management [2, 5].

The pathophysiologic mechanisms of many chronic pain syndromes are thought to involve central sensitization (CS) [6], which develops when nociceptor inputs increase the excitability of neurons in central nociceptive pathways [7]. CS generally manifests as allodynia (perceiving pain in response to a stimulus that does not normally induce pain), hyperalgesia (an elevated sensitivity to a normally painful stimulus), and aftersensations following cessation of the pain stimulus [7].

CS may be assessed through quantitative sensory testing, which evaluates the responses to thermal and mechanical stimuli [8]. Simpler methods of assessing CS include questionnaires such as the Central Sensitization Inventory (CSI), which is a patient-reported instrument designed to screen populations at risk of CS or to assess physical and emotional symptoms associated with CS [9]. The 25-item CSI (CSI-25) consists of two sections: part A rates the frequency of occurrence of 25 common symptoms of CS using a 5-point Likert scale (never, rarely, sometimes, often, or always), while part B is not scored but is used to report the previous diagnosis of 10 diseases related to central sensitivity syndrome (CSS). The CSI-25 has been shown to distinguish between patients with chronic pain and controls. However, the availability of a shorter form of the CSI would reduce the burden on the patient and thereby facilitate the screening of CS in both clinical and research settings.

In 2018, Nishigami et al. used successive Rasch analyses to simplify the CSI-25 into a 9-item CSI (CSI-9) [10]. The CSI-9 rates the frequency of occurrence of nine common symptoms of CS using a 5-point Likert scale (never, rarely, sometimes, often, or always), with a higher score indicating more severe CS. The CSI-9 was shown to have acceptable psychometric properties [10]. The main advantage of the CSI-9 is its practicality in clinical practice since it involves only nine items and is fast and easy to use. Furthermore, an optimal cut-off value of 20 points was shown to distinguish patients with fibromyalgia from those with musculoskeletal pain, while an optimal cut-off value of 17 points was demonstrated to distinguish patients with fibromyalgia from a healthy population of research subjects [11]. The above study suggested that a CSI-9 score of >20 could be used to screen for CS and help physicians identify patients who need targeted treatment. Research has shown that pain can be influenced by culture and that there are variations in pain beliefs, pain assessments, pain treatment strategies, and catastrophic pain scores between different countries and different languages [12–14]. Cultural differences can lead to bias, so it is essential to adopt a Chinese version for the application of CSI-9 in China. With the convenience of CSI-9 in clinical use, a Chinese CSI-9 would definitely improve the current situation in China. Still, no studies have described the translation and cross-cultural adaptation of the CSI-9 into Chinese, but a recent study reported a Chinese version of the CSI-25 [15].

The aim of this single-center study was to cross-culturally adapt the CSI-9 to Chinese and assess its psychometric characteristics (including internal consistency, test-retest reliability, construct validity, and factor structure) in patients with chronic pain. It was anticipated that the Chinese version of the CSI-9 would provide clinicians and researchers with a new instrument for the evaluation of chronic pain and CS.

## Methods

### Study participants

This single-center, cross-sectional study enrolled patients with chronic pain who attended the outpatient department or were admitted to the ward of the Rheumatology and Immunology

Department, the First Medical Center, the Chinese People's Liberation Army General Hospital. A group of healthy controls was recruited from the physical examination center of the First Medical Center, the Chinese People's Liberation Army General Hospital, during the same period. The physical examination center, a department localized inside the hospital, offers routine regular physical examinations for ordinary people. The Ethics Committee of the Chinese People's Liberation Army General Hospital approved this study (No. S2021-285-02), and all participants provided written consent.

The inclusion criteria for the patients with chronic pain were: (1) male or female aged >18 years old; (2) diagnosed with chronic pain (the presence of pain for ≥3 months), including fibromyalgia or musculoskeletal pain (e.g., lumbago, cervicodynia, hip pain, knee pain, ankle pain, shoulder pain, elbow pain, hand and wrist pain, lateral epicondylalgia, and temporomandibular joint pain); (3) pain severity, scored on the Numeric Rating Scale (NRS) of 0–10, was ≥3 points for the majority of time during the previous 1 week; and (4) pain symptoms and treatment regimen had been stable for >1 month. The inclusion criteria for the healthy controls were: (1) male or female aged >18 years old; (2) not diagnosed with CS or chronic pain during the previous 5 years; and (3) no long-term pain complaints. Patients with chronic pain were excluded from the analysis if any of the following criteria were met: (1) history of trauma or fracture during the previous 6 weeks; (2) acute disease (e.g., acute infection) during the previous 4 weeks; (3) cancer; (4) brain or spinal cord injury; (5) neurologic disease; (6) serious underlying disease (such as severe cardiopulmonary, gastrointestinal or genitourinary disease) that might affect the scoring of the scale; and (7) mental illness or severe emotional disorder. Additional exclusion criteria for both groups were: (1) the participant had difficulty interpreting Chinese or had a reading disorder; (2) in the opinion of the researchers, the participant would not be able to fully cooperate with the study protocol (including completion of the questionnaires) or had difficulty communicating; (3) the participant did not fill in all the items in the CSI-9 scale; (4) the participant consistently chose a particular option or exhibited clear regularity in the selection of answers; and (5) the participant failed to answer the questions in accordance with the instructions or provided unrelated answers.

## Study design

The study was divided into two parts. First, the original CSI-9 was translated, back-translated, and cross-culturally adapted using the Brislin bidirectional translation method [16]. Second, the sociodemographic data of patients and healthy controls were collected, and the participants were asked to fill in the Chinese CSI-9, Brief Pain Inventory (BPI), and Pain Catastrophic Scale (PCS) on site.

## Development of the Chinese CSI-9

The study researchers contacted the author of the original CSI-9, Katsuyoshi Tanaka, by email in May 2021 and were given permission to develop a Chinese version of the scale. The process of translation, back-translation, and cross-cultural adaptation was carried out in strict accordance with established guidelines to ensure maximum equivalence between the Chinese version of the CSI-9 and the original instrument [16]. Seven experts in fibromyalgia, pain, and the psychology of pain (all working in pain or rheumatology & immunology departments) participated in the Delphi consultation on the Chinese version. With regard to the translation process, the implementation process was as follows. A preliminary Chinese version of the scale was generated by translating and back-translating the original CSI-9 using the Brislin bidirectional translation method. First, two experts (LD and ZG) and a Chinese professional translator (Shanghai Richard Translation Co.) translated the scale from English to Chinese, and the

researchers integrated it into the original Chinese version. Then, a professional translator of Chinese nationality and a professional English native-speaking translator (Shanghai Richard Translation Co.) back-translated the original Chinese version of the scale, namely, Chinese to English. After that, the researchers combined the results of the English-to-Chinese translation and Chinese-to-English translation to form the first Chinese version of the scale again. Finally, the first Chinese version of the scale was tested by doctors and patients, and the translated items were modified again according to the test results to obtain the final version of the Chinese CSI-9.

Six patients with chronic pain who had different ages, genders, and education levels were asked to provide feedback on version B of the Chinese CSI-9 as part of a pilot test. The respondents gave their opinions on the comprehensibility and accuracy of the questions and answers and provided an overall evaluation of the questionnaire. Additionally, the response time and response rate were recorded. Next, a group of seven experts in the diagnosis and treatment of chronic pain and six patients used the Delphi method [17] to revise and optimize the wording and structure of the items in the scale after taking into account the opinions and feedback of the patients, thereby generating the final version of the Chinese CSI-9.

The final version of the Chinese CSI-9 comprised two parts. Part A contained 9 items, each of which was scored using a 5-point Likert scale with "0" for "never" and "4" for "always". The total score of part A ranged from 0 to 36, and a higher score indicated more severe CS. Severity was divided into three grades: subclinical (0–9 points), mild (10–19 points), and moderate/severe ($\geq$20 points) [10]. Part B was not scored but was used to obtain information regarding a history of CS-related diseases, including restless leg syndrome (RLS), chronic fatigue syndrome (CFS), fibromyalgia, TMD, migraine or TTH, IBS, multiple chemical sensitivity (MCS), neck injury (including whiplash injury), anxiety or panic attacks, and depression.

## Data collection

All participants were asked to fill in paper versions of the Chinese CSI-9 and two comparator scales (the Chinese versions of the BPI [18] and PCS [19] scales) on-site. The comparator scales were used to investigate the construct validity of the Chinese CSI-9. The Chinese BPI assesses the characteristics of pain, such as intensity, location, and duration. The Chinese PCS contains 13 items graded on a 5-point Likert scale ("0" for "never" and "4" for "always"), and its total score ranges from 0 to 52, with a higher score indicative of more catastrophic pain. Sociodemographic data were also obtained for all the participants.

## Validity assessment

Confirmatory factor analysis was used to assess structural validity by evaluating the similarity of the dimensions and factor loadings between the Chinese CSI-9 and the original CSI-9. The indices used to determine the model fit were: chi-square/degrees of freedom ($\chi^2$/df), goodness-of-fit index (GFI), adjusted goodness-of-fit index (AGFI), comparative fit index (CFI), Tucker-Lewis Coefficient (TLI) and root mean square error of approximation (RMSEA). A model with $\chi^2$/df<3, RMSEA<0.08, GFI>0.90, CFI>0.90, and TLI>0.90 suggested a good fit [20].

Construct validity was assessed by examining the correlations between the Chinese CSI-9 score and the Chinese PCS total score, Chinese PCS subscale scores (rumination, magnification, and helplessness), Chinese BPI total score, Chinese BPI mean item score, pain duration, and the number of body locations experiencing pain. Each correlation was measured by determining Spearman's rank correlation coefficient ($r$).

Criterion validity was explored by comparing the Chinese CSI-9 part A score between patients with/without each CSS-related diagnosis in part B and between patients with/without one CSS-related diagnosis, two CSS-related diagnoses, and more than three CSS-related diagnoses according to part B. Discriminatory analysis was also used to compare the differences in each item between patients with chronic pain and healthy controls. The criteria for interpreting model fit and magnitude of correlations in convergent validity were specified in advance.

## Reliability assessment

Cronbach's α coefficient was used to evaluate internal consistency [21]. In this study, reliability was poor for $0.5 \leq$ Cronbach's $\alpha < 0.6$, acceptable for $0.6 \leq$ Cronbach's $\alpha < 0.7$, good for $0.7 \leq$ Cronbach's $\alpha < 0.9$, and excellent for Cronbach's $\alpha \geq 0.9$.

Test-retest reliability was used to assess the stability of the CSI-9. When tested for the first time, all subjects were confirmed by randomization whether to be tested for the second time at $7 \pm 1$ days after the first test. Those who were retested were required to maintain a stable therapeutic regimen (i.e., no changes in drugs and dosages) between the first and second tests. The intraclass correlation coefficient (ICC) was calculated by a two-way random-effects model; an absolute agreement was used, and test-retest reliability was classified as moderate for $0.50 \leq ICC < 0.75$, good for $0.75 \leq ICC < 0.90$, and excellent for $ICC \geq 0.90$ [22].

## Measurement error

Bland-Altman plots were generated to determine the mean differences and visualize systematic errors in the baseline.

## Floor and ceiling effects

Floor and ceiling effects were considered to be present if $\geq 15\%$ of the patients reported the lowest (0) or highest (100) possible CSI score [23].

## Exploratory analysis of the utility of the Chinese CSI-9 as a screening tool for CS

Receiver operating characteristic (ROC) curve analyses were used to evaluate whether the Chinese CSI-9 might have utility as a screening tool for CS. Optimal cut-off values for the CSI-9 score were determined according to the Youden index. The area under the ROC curve (AUC) and its 95% confidence interval (95%CI), sensitivity, specificity, positive predictive value (PPV), negative predictive value (PPV), and accuracy were calculated.

## Statistical analysis

The sample size calculation was based on CFA, as designed in the protocol, and including at least 200 patients would meet the needs for validation, and 50 would be sufficient for retest reliability. The analyses were performed using SPSS 22.0 and AMOS 23.0 (IBM, Armonk, NY, USA), and the Bland-Altman plots were drawn using MedCalc 19.6.1 (MedCalc Software, Ostend, Belgium). All statistical tests were two-sided, and $P < 0.05$ was considered statistically significant. Continuous variables conforming to a normal distribution are shown as the mean ± standard deviation, and those not conforming to a normal distribution are expressed as the median (interquartile range). Categorical variables are expressed as the number of cases (percentage). Continuous variables conforming to a normal or approximately normal distribution were compared between two groups using the t-test for independent samples and among multiple groups using one-way analysis of variance (ANOVA). Non-normally distributed

continuous variables were compared between groups using the Mann-Whitney U test (two groups) or the Kruskal-Wallis test (multiple groups). The chi-squared test or Fisher's exact test was used to analyze categorical data.

## Results

### Cross-cultural adaptation of the CSI-9 into Chinese

The CSI was forward-translated into Chinese and backward-translated into English without difficulty. The patients in the pilot test indicated that the Chinese CSI-9 was straightforward to understand, so no further changes were made to the scale.

### Baseline characteristics of the study participants

A total of 308 participants were screened for inclusion in the study. Two participants were excluded for repeated entry of information, eleven were excluded for failing to follow the instructions on the questionnaires or failing to complete the questionnaires accurately/fully, and five were excluded because the pain had been present for less than 3 months. Therefore, 290 participants were included in the final analysis.

The baseline characteristics of the study participants are presented in Table 1. The study population included 235 patients with chronic pain (114 cases of fibromyalgia and 121 cases of musculoskeletal pain) and 55 healthy controls. There were significant differences between groups in age, gender, body mass index (BMI), and employment status ($P < 0.05$) but not weight, height, marital status, or years of education (Table 1). Furthermore, the pain severity score, PCS score, PCS subscale scores, CSI-9 score, and CSI severity score were all markedly higher in patients with chronic pain than in healthy controls ($P < 0.001$; Table 1).

### Structural validity

Table 2 summarizes the results of the structural validity evaluation (Table 2). The model fit indices were $\chi^2$ = 97.608, df = 27, $\chi^2$/df = 3.615, TLI = 0.927, CFI = 0.945, RMSEA = 0.095, GFI = 0.929, AGFI = 0.882. The result of the correlations met the pre-specified criteria for convergent and divergent validities of the Chinese CSI-9 with those variables.

### Construct validity

As shown in Table 3, the calculation of the Pearson correlation coefficient demonstrated that the CSI-9 score was positively correlated with the PCS total score ($r$ = 0.463), PCS rumination, magnification, and helplessness subscale scores ($r$ = 0.347–0.463), BPI mean item score ($r$ = 0.524), BPI total score ($r$ = 0.773), and the number of painful body sites ($r$ = 0.451). However, the CSI-9 score was not correlated with the duration of pain (Table 3).

### Test-retest reliability

Among 117 patients with chronic pain who completed the CSI-9 on two separate occasions, three were excluded because the interval between the two tests was less than 6 days, three were excluded because the questionnaires were incomplete or clearly contained errors, and two were excluded because they had suffered from chronic pain for <90 days. Therefore, 109 patients were included in the assessment of test-retest reliability. The test-retest reliability (two-way random-effects model) of the CSI-9 total score (Table 4) was excellent in the overall sample (ICC = 0.958) and was good in the chronic pain group (ICC = 0.875) and healthy control group (ICC = 0.807). Furthermore, the test-retest reliability was good-to-excellent

**Table 1. Baseline data.**

| Variable | Overall sample (n = 290) | Fibromyalgia (n = 114) | Musculoskeletal pain (n = 121) | Healthy controls (n = 55) | P |
|---|---|---|---|---|---|
| Age (years) | 44.0 ± 12.7 | 41.6 ± 12.0 | 46.0 ± 13.9 | 44.8 ± 10.3 | 0.026 |
| Gender | | | | | 0.021 |
| Male | 68 (23.4%) | 17 (14.9%) | 36 (29.8%) | 15 (27.3%) | |
| Female | 222 (76.6%) | 97 (85.1%) | 85 (70.2%) | 40 (72.7%) | |
| Weight (kg) | 62.9 ± 10.5 | 61.5 ± 10.0 | 63.1 ± 10.8 | 65.5 ± 10.7 | 0.066 |
| Height (cm) | 163.6 ± 7.4 | 163.3 ± 7.1 | 164.6 ± 7.7 | 162.3 ± 7.0 | 0.139 |
| BMI (kg/m$^2$) | 23.5 ± 3.4 | 23.0 ± 3.3 | 23.3 ± 3.2 | 24.8 ± 3.5 | 0.006 |
| Marital status | | | | | 0.849 |
| Married | 243 (83.8%) | 94 (82.5%) | 102 (84.3%) | 47 (85.5%) | |
| Single | 32 (11.0%) | 15 (13.2%) | 11 (8.9%) | 6 (10.9%) | |
| Divorced | 10 (3.4%) | 3 (2.6%) | 5 (4.1%) | 2 (3.6%) | |
| Widowed | 5 (1.7%) | 2 (1.8%) | 3 (2.4%) | 0 | |
| Education level (years) | 11.7 ± 4.1 | 12.3 ± 4.0 | 11.5 ± 4.4 | 11.0 ± 3.2 | 0.090 |
| Employment status | | | | | <0.001 |
| Unemployed | 91 (31.2%) | 42 (36.8%) | 40 (32.5%) | 9 (16.4%) | |
| Employed | 148 (51.0%) | 49 (43.0%) | 56 (46.3%) | 43 (78.2%) | |
| Retired | 44 (15.2%) | 18 (15.8%) | 24 (19.8%) | 2 (3.6%) | |
| Student | 7 (2.4%) | 5 (4.4%) | 1 (0.8%) | 1 (1.8%) | |
| BPI score | 42.3 ± 27.0 | 58.8 ± 18.6 | 45.6 ± 19.1 | 0.4 ± 1.7 | <0.001 |
| Pain duration (months) | 2042.4 ± 2697.4 | 1970.4 ± 2786.6 | 2110.2 ± 2620.3 | 0.0 ± 0.0 | 0.693 |
| Location of pain | | | | | |
| Head | 81 (27.9%) | 56 (49.1%) | 24 (19.8%) | 1 (1.8%) | <0.001 |
| Neck | 135 (46.6%) | 79 (69.3%) | 55 (45.5%) | 1 (1.8%) | <0.001 |
| Chest | 49 (16.9%) | 37 (32.5%) | 12 (9.9%) | 0 | <0.001 |
| Back | 147 (50.7%) | 85 (74.6%) | 62 (51.2%) | 0 | <0.001 |
| Abdomen | 35 (12.1%) | 30 (26.3%) | 5 (4.1%) | 0 | <0.001 |
| Waist | 151 (52.1%) | 79 (69.3%) | 71 (58.7%) | 1 (1.8%) | <0.001 |
| Hip | 107 (36.9%) | 70 (61.4%) | 37 (30.6%) | 0 | <0.001 |
| Left shoulder | 122 (42.1%) | 78 (68.4%) | 44 (36.4%) | 0 | <0.001 |
| Left upper arm | 83 (28.6%) | 61 (53.5%) | 22 (18.2%) | 0 | <0.001 |
| Left elbow | 69 (23.8%) | 51 (44.7%) | 18 (14.9%) | 0 | <0.001 |
| Left forearm | 57 (19.7%) | 43 (37.7%) | 14 (11.6%) | 0 | <0.001 |
| Left palm | 70 (24.1%) | 43 (37.7%) | 27 (22.3%) | 0 | <0.001 |
| Left thigh | 73 (25.2%) | 52 (45.6%) | 21 (17.4%) | 0 | <0.001 |
| Left knee | 118 (40.7%) | 69 (60.5%) | 49 (40.5%) | 0 | <0.001 |
| Left lower leg | 68 (23.4%) | 53 (46.5%) | 13 (10.7%) | 2 (3.6%) | <0.001 |
| Left foot | 78 (26.9%) | 55 (48.2%) | 23 (19.0%) | 0 | <0.001 |
| Right shoulder | 123 (42.4%) | 78 (68.4%) | 45 (37.2%) | 0 | <0.001 |
| Right upper arm | 82 (28.3%) | 63 (55.3%) | 19 (15.7%) | 0 | <0.001 |
| Right elbow | 72 (24.8%) | 56 (49.1%) | 16 (13.2%) | 0 | <0.001 |
| Right forearm | 65 (22.4%) | 51 (44.7%) | 14 (11.6%) | 0 | <0.001 |
| Right palm | 74 (25.5%) | 41 (36.0%) | 32 (26.4%) | 1 (1.8%) | <0.001 |
| Right thigh | 79 (27.2%) | 58 (50.9%) | 21 (17.4%) | 0 | <0.001 |
| Right knee | 129 (44.5%) | 73 (64.0%) | 56 (46.3%) | 0 | <0.001 |
| Right lower leg | 72 (24.8%) | 59 (51.8%) | 13 (10.7%) | 0 | <0.001 |
| Right foot | 80 (27.6%) | 54 (47.4%) | 26 (21.5%) | 0 | <0.001 |

*(Continued)*

**Table 1.** (Continued)

| Variable | Overall sample | Fibromyalgia | Musculoskeletal pain (n | Healthy controls | P |
|---|---|---|---|---|---|
| | (*n* = 290) | (*n* = 114) | (*n* = 121) | (*n* = 55) | |
| **PCS score** | 24.7 ± 14.40 | 32.86 ± 9.93 | 25.60 ± 12.50 | 5.85 ± 7.18 | <0.001 |
| **PCS subscale scores** | | | | | |
| Rumination | 9.0 ± 4.8 | 11.3 ± 3.5 | 9.7 ± 4.1 | 2.8 ± 3.3 | <0.001 |
| Magnification | 5.2 ± 3.5 | 6.9 ± 2.6 | 5.4 ± 3.3 | 1.1 ± 1.5 | <0.001 |
| Helplessness | 10.7 ± 7.0 | 14.7 ± 5.2 | 10.8 ± 6.2 | 2.2 ± 3.3 | <0.001 |
| **CSI score** | 18.0 ± 9.0 | 25.2 ± 5.3 | 17.0 ± 6.2 | 5.1 ± 3.7 | <0.001 |
| **CSI severity** | | | | | <0.001 |
| Subclinical (0 to 9) | 62 (21.4%) | 0 (0.0%) | 15 (12.4%) | 47 (85.5%) | |
| **Mild (10 to 19)** | 87 (30.0%) | 14 (12.3%) | 65 (53.7%) | 8 (14.5%) | |
| Moderate/severe (≥20) | 141 (48.6%) | 100 (87.7%) | 41 (33.9%) | 0 (0.0%) | |
| **Number of diagnoses in CSI part B** | 0.9 ± 1.1 | 1.8 ± 1.1 | 0.6 ± 0.9 | 0.0±0.0 | |

(ICC ≥ 0.75) for all of the items in the overall sample, 6 of the 9 items in the chronic pain group, but only 2 of the 9 items in the healthy control group (Table 4).

## CSS-related disorders

The CSS-related disorders reported by the patients in part B of the CSI-9 are shown in the S1 Table. The average number of diagnoses for patients with chronic pain was 1.17 ± 1.14, and the proportion of patients reporting 1, 2, and ≥3 CSS-related syndromes was 38.3%, 16.6%, and 13.2%, respectively. The most common CSS-related disorders reported by the patients with chronic pain were fibromyalgia (51.5%), anxiety/panic attacks (20.9%), migraine/tension-type headache (TTH) (17.0%), and depression (12.8%). The remaining six CSS-related syndromes were each reported by less than 5% of the patients with chronic pain. None of the healthy controls reported any CSS-related disorders.

## Measurement error

Bland-Altman plots revealed that the mean differences for the overall sample and patients with chronic pain were not significantly different from zero, and no systematic bias was detected (Fig 1).

## Criterion validity

Table 5 shows that the CSI-9 score was significantly higher in patients with at least one CSS-related diagnosis in part B of the instrument (*P* < 0.001 vs. no CSS-related diagnoses), patients

**Table 2. Structural validity of the Chinese central sensitization inventory.**

| No. | Item | Mean ± SD | Factor 1 |
|---|---|---|---|
| 1 | Unrefreshed in morning | 2.2 ± 1.3 | 0.692 |
| 2 | Muscles stiff/achy | 2.5 ± 1.4 | 0.773 |
| 3 | Pain all over body | 2.2 ± 1.5 | 0.865 |
| 4 | Headaches | 1.3 ± 1.2 | 0.567 |
| 5 | Do not sleep well | 2.1 ± 1.4 | 0.711 |
| 6 | Difficulty concentrating | 1.6 ± 1.3 | 0.712 |
| 7 | Stress makes symptoms worse | 1.5 ± 1.3 | 0.692 |
| 8 | Tension neck and shoulder | 2.3 ± 1.4 | 0.752 |
| 9 | Poor memory | 2.1 ± 1.3 | 0.623 |

**Table 3. Construct validity in the overall sample ($n$ = 290).**

| Scale | Construct validity | $r$ | 95% CI |
|---|---|---|---|
| CSI | PCS | 0.463 | 0.354, 0.557 |
| | Rumination subscale of PCS | 0.347 | 0.225, 0.467 |
| | Magnification subscale of PCS | 0.386 | 0.275, 0.492 |
| | Helplessness subscale of PCS | 0.463 | 0.361, 0.568 |
| | BPI mean item score | 0.524 | 0.421, 0.616 |
| | BPI total score | 0.773 | 0.416, 0.620 |
| | Duration of pain | -0.003 | -0.133, 0.126 |
| | Number of sites of pain | 0.451 | 0.353, 0.547 |

BPI: Brief Pain Inventory; CI: confidence interval; CSI: Central Sensitization Inventory; PCS: Pain Catastrophic Scale; r: Pearson correlation coefficient.

with at least two CSS-related diagnoses ($P < 0.001$ vs. ≤1 CSS-related diagnosis) and patients with at least 3 CSS-related diagnoses ($P < 0.001$ vs. ≤2 CSS-related diagnoses). When the items in part B were analyzed individually, the CSI-9 score was significantly higher in patients who had fibromyalgia ($P < 0.001$), migraine/TTH ($P < 0.001$), IBS ($P = 0.040$), neck injury ($P = 0.021$), anxiety/panic attacks ($P < 0.001$) or depression ($P < 0.001$) when compared with patients who did not report these CSS-related disorders (Table 5). The CSI-9 score did not differ significantly between patients with/without TMD, RLS, CFS, or MCS (Table 5).

## Internal consistency

For the overall sample, Cronbach's α value was 0.902 for the overall scale, indicating excellent internal consistency (Table 6). For patients with chronic pain, Cronbach's α value was 0.828 for the overall scale, with good internal consistency (Table 6). The Cronbach's alpha displayed in Table 6 for each item represented the total Cronbach's alpha of CSI-9 except the corresponding item.

**Table 4. Test-retest reliability (two-way random-effects model).**

| | | Overall sample | | Chronic pain | | Healthy controls | |
|---|---|---|---|---|---|---|---|
| | | ($n$ = 109) | | ($n$ = 58) | | ($n$ = 51) | |
| No. | Items | ICC | 95% CI | ICC | 95% CI | ICC | 95% CI |
| 1 | Unrefreshed in morning | 0.798 | 0.718, 0.857 | 0.678 | 0.510, 0.796 | 0.639 | 0.445, 0.776 |
| 2 | Muscles stiff/achy | 0.939 | 0.912, 0.958 | 0.705 | 0.547, 0.814 | 0.807 | 0.684, 0.885 |
| 3 | Pain all over body | 0.924 | 0.891, 0.947 | 0.803 | 0.688, 0.879 | 0.563 | 0.343, 0.725 |
| 4 | Headaches | 0.815 | 0.741, 0.870 | 0.797 | 0.679, 0.874 | 0.422 | 0.168, 0.624 |
| 5 | Do not sleep well | 0.837 | 0.771, 0.886 | 0.842 | 0.747, 0.903 | 0.479 | 0.235, 0.666 |
| 6 | Difficulty concentrating | 0.851 | 0.789, 0.895 | 0.804 | 0.690, 0.879 | 0.717 | 0.552, 0.828 |
| 7 | Stress makes symptoms worse | 0.813 | 0.738, 0.868 | 0.739 | 0.596, 0.837 | 0.633 | 0.434, 0.773 |
| 8 | Tension neck and shoulder | 0.884 | 0.834, 0.920 | 0.805 | 0.692, 0.879 | 0.710 | 0.541, 0.823 |
| 9 | Poor memory | 0.866 | 0.810, 0.906 | 0.865 | 0.783, 0.918 | 0.609 | 0.404, 0.756 |
| All | Total score | 0.958 | 0.939, 0.971 | 0.875 | 0.798, 0.924 | 0.807 | 0.684, 0.885 |

ICC, intraclass correlation coefficient; CI, confidence interval.

The ICC values shown for each item resulted from the remaining items after the deletion of that item.

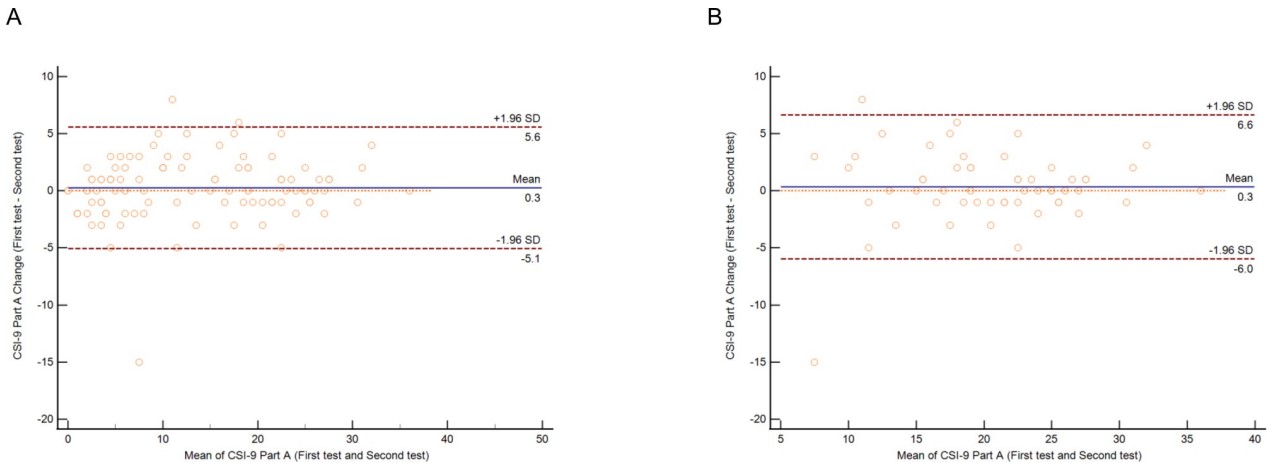

**Fig 1. Bland-Altman plots for evaluating measurement error.** A. Overall sample. B: Patients with chronic pain. The mean differences for the overall sample and chronic pain group did not significantly differ from zero, and no systematic bias was detected.

## Discriminatory analysis

The patients with chronic pain had a higher score for every item of part A of the CSI-9 than the group of healthy controls ($P < 0.001$ for all items) (S2 Table).

**Table 5. Criterion validity in the overall sample (n = 290).**

| No. | CSS diagnosis (CSI part B) | Presence of symptom | n (%) | CSI part A score (mean ± SD) | P |
|---|---|---|---|---|---|
| 1 | Restless leg syndrome | Yes | 4 (1.4%) | 22.3 ± 7.5 | 0.278 |
| | | No | 286 (98.6%) | 17.9 ± 9.1 | |
| 2 | Chronic fatigue syndrome | Yes | 2 (0.7%) | 21.5 ± 6.4 | 0.603 |
| | | No | 288 (99.3%) | 17.9 ± 9.1 | |
| 3 | Fibromyalgia | Yes | 121 (41.7%) | 24.8 ± 5.6 | <0.001 |
| | | No | 169 (58.3%) | 13.1 ± 7.8 | |
| 4 | Temporomandibular joint disorder | Yes | 4 (1.4%) | 24.3 ± 5.0 | 0.160 |
| | | No | 286 (98.6%) | 17.9 ± 9.1 | |
| 5 | Migraine or tension headache | Yes | 40 (13.8%) | 24.2 ± 6.4 | <0.001 |
| | | No | 250 (86.2%) | 16.9 ± 9.0 | |
| 6 | Irritable bowel syndrome | Yes | 8 (2.8%) | 24.5 ± 5.7 | 0.040 |
| | | No | 282 (97.2%) | 17.8 ± 9.1 | |
| 7 | Multiple chemical sensitivities | Yes | 5 (1.7%) | 22.8 ± 7.9 | 0.204 |
| | | No | 285 (98.3%) | 17.9 ± 9.0 | |
| 8 | Neck injury (including whiplash) | Yes | 11 (3.8%) | 23.8 ± 5.4 | 0.021 |
| | | No | 279 (96.2%) | 17.7 ± 9.1 | |
| 9 | Anxiety or panic attacks | Yes | 49 (16.9%) | 16.7 ± 9.1 | <0.001 |
| | | No | 241 (83.1%) | 24.2 ± 5.5 | |
| 10 | Depression | Yes | 30 (10.3%) | 25.4 ± 6.1 | <0.001 |
| | | No | 260 (89.7%) | 17.1 ± 8.9 | |
| | At least 1 CSS symptom | Yes | 90 (31.0%) | 21.6 ± 6.1 | <0.001 |
| | | No | 200 (69.0%) | 16.3 ± 9.7 | |
| | At least 2 CSS symptoms | Yes | 39 (13.4%) | 23.8 ± 5.2 | <0.001 |
| | | No | 251 (86.6%) | 17.0 ± 9.2 | |
| | At least 3 CSS symptoms | Yes | 31 (10.7%) | 27.4 ± 4.5 | <0.001 |
| | | No | 259 (89.3%) | 16.8 ± 8.8 | |

**Table 6. Internal consistency.**

| No. | CSI items | Overall sample (N = 290) | Chronic pain sample (N = 235) |
|---|---|---|---|
| 1 | Unrefreshed in the morning | 0.893 | 0.811 |
| 2 | Muscles stiff/achy | 0.888 | 0.819 |
| 3 | Pain all over body | 0.880 | 0.794 |
| 4 | Headache. | 0.901 | 0.824 |
| 5 | 'Do not sleep well | 0.891 | 0.807 |
| 6 | Difficulty concentrating | 0.890 | 0.802 |
| 7 | Stress makes symptoms worse | 0.892 | 0.806 |
| 8 | Tension neck and shoulder | 0.889 | 0.808 |
| 9 | Poor memory | 0.897 | 0.824 |
|  | Total | 0.902 | 0.828 |

The ICC values shown for each item resulted from the remaining items after the deletion of that item.

## Floor and ceiling effects

Eight participants (2.8%) had a CSI-9 score of 0, and two participants (0.7%) had a CSI-9 score of 100 points. Therefore, ceiling and floor effects were not observed.

## Exploratory analysis of the utility of the Chinese CSI-9 as a screening tool for CS

ROC curve analysis demonstrated that a CSI-9 score > 18 points (the optimal cut-off value) predicted the presence of CS in the overall sample with an AUC of 0.873 (95%CI, 0.829–0.909), a sensitivity of 79.4%, a specificity of 79.2%, a PPV of 82.5%, an NPV of 75.7% and an accuracy of 79.3% (Table 7). Interestingly, the sensitivity, NPV, and accuracy were lower when a cut-off value of 20 points was used, as described previously [11], although the specificity and PPV were higher using a cut-off value of 20 points (Table 7). In the chronic pain group, a CSI-9 value > 19 points (the optimal cut-off) detected the presence of CS with an AUC of 0.785 (95%CI, 0.727–0.836), a sensitivity of 74.4%, a specificity of 70.7%, a PPV of 84.4%, an NPV of 56.4% and an accuracy of 73.2% (Table 7).

## Discussion

This study successfully developed a new version of the CSI-9 intended for use in China through a process that involved translation, back-translation, and cultural adaptation. The CSI-9 had reasonable construct validity, good-to-excellent test-retest reliability, good criterion validity, and excellent internal consistency. Furthermore, confirmatory factor analysis was used to test the structural validity, with one main factor that explained the majority of the total

**Table 7. Utility of the Chinese 9-item Central Sensitization Inventory in the diagnosis of central sensitization.**

| Population | Outcome[a] | Cut-off | Sensitivity (95%CI) | Specificity (95%CI) | PPV | NPV | Accuracy | AUC (95%CI) |
|---|---|---|---|---|---|---|---|---|
| **Overall sample (_n_ = 290)** | Presence of CS | 18 | 79.4% (72.3%, 85.4%) | 79.2% (71.2%, 85.8%) | 82.5% | 75.7% | 79.3% | 0.873 (0.829, 0.909) |
| **Overall sample (_n_ = 290)** | | 20[b] | 69.4% (61.6%, 76.4%) | 85.4% (78.1%, 91.0%) | 85.4% | 69.4% | 76.6% | — |
| **Chronic pain sample (_n_ = 235)** | | 19 | 74.4% (66.9%, 80.9%) | 70.7% (59.0%, 80.6%) | 84.4% | 56.4% | 73.2% | 0.785 (0.727, 0.836) |

[a] Persons who had any one of the diseases in Part B of the CSI were diagnosed as having central sensitization. [b] A cut-off value of 20 was used according to the literature;[15] the optimal cut-off values of 18 and 19 were calculated in the present study. AUC: area under the receiver operating characteristic curve; CI: confidence interval; CS: central sensitization; NPV: negative predictive value; PPV: positive predictive value.

variance for the items in part A of the Chinese CSI-9. The instrument developed in this study could potentially be utilized as a tool to screen for CS in patients with chronic pain in China.

Structural validity relates to the extent to which the scores of a scale are an adequate reflection of the dimensionality of the construct to be measured [24]. Confirmatory factor analysis identified one factor between the items in part A of the Chinese CSI-9, along with the original CSI-9 [10]. Nevertheless, the factors have been reported for various versions of the CSI-25. For example, confirmatory factor analysis of the original English language version of the CSI-25 identified four main factors ("physical symptoms", "emotional distress", "headache/jaw symptoms", and "urological symptoms") [9], and the same four factors were described for the German CSI-25 [25]. Four main factors were also reported for the Dutch CSI-25 ("general disability and physical symptoms", "emotional distress", "higher central sensitivity", and "urological and dermatological symptoms") [26], while the Japanese version identified five factors ("emotional distress", "headache/jaw symptoms", "urological and general symptoms", "muscle symptoms" and "sleep disturbance") [27].

The construct validity of the Chinese CSI-9 was assessed by comparing it with other scales (the PCS and BPI) that evaluate similar qualities. The Chinese CSI-9 score was positively correlated with the total PCS score ($r = 0.463$) and its rumination, magnification, and helplessness subscale scores ($r = 0.347–0.463$). Furthermore, the Chinese CSI-9 score was also positively correlated with the BPI mean item score ($r = 0.524$), BPI total score ($r = 0.773$), and the number of painful body sites ($r = 0.451$). Hence, the Chinese CSI-9 exhibited a good correlation with a scale that evaluates pain characteristics such as intensity, location, and duration (BPI) but was less strongly correlated with a scale that assesses an individual's experience of pain (PCS). Published data are lacking regarding the correlation of the original CSI-9 with other instruments, such as the BPI and PCS. However, our findings are broadly consistent with previous studies exploring the construct validity of the CSI-25. For example, the Greek CSI-25 correlated with the PCS score ($r = 0.680$) [28], and the Nepali CSI-25 correlated with the PCS score ($r = 0.50$), pain intensity measured by the Numerical Rating Scale (NRS; $r = 0.25$), and the total number of pain types ($r = 0.35$) [29]. The Japanese CSI-25 was positively correlated with the pain intensity ($r = 0.42$) and pain interference ($r = 0.48$) scores of the BPI [27], and the Italian CSI-25 was correlated with the NRS score ($r = 0.427$) [30]. The Chinese CSI-9 score was not correlated with the duration of pain, which is consistent with previous analyses of the CSI-25 [27, 29].

The test-retest reliability of the Chinese CSI-25 was excellent in the overall sample (ICC = 0.958), which is consistent with the good test-retest reliability reported for the original CSI-9 (ICC = 0.79) [10]. The test-retest reliability of the CSI-9 was also comparable to that described for the English (ICC = 0.817) [9], German (ICC = 0.917) [25], Dutch (ICC = 0.88–0.91) [26], Japanese (ICC = 0.85) [27], Greek (ICC = 0.991) [28], Nepali (ICC = 0.98) [29] and Persian (ICC = 0.934) [31] versions of the CSI-25.

The Chinese CSI-9 had a Cronbach's α value of 0.902 in the overall sample (excellent internal consistency) and 0.828 in patients with chronic pain (good internal consistency), which compares with a Cronbach's α value of 0.80 for the original CSI-9 [10], 0.879 for the English CSI-25 [9], 0.928 for the German CSI-25 [25], 0.78 for the Dutch CSI-25 [26], 0.89 for the Japanese CSI-25 [27], 0.993 for the Greek CSI-25 [28], 0.91 for the Nepali CSI-25 [29], 0.87 for the Italian CSI-25 [30], 0.87 for the Persian CSI-25 [31] and 0.872 for the Spanish CSI-25 [32].

A prior analysis of the original CSI-9 found that a cut-off value of 20 points was optimal for distinguishing patients with fibromyalgia from patients with musculoskeletal disorders (sensitivity of 92.3% and specificity of 93.3%), while a cut-off value of 17 points was optimal for distinguishing patients with fibromyalgia from healthy controls (sensitivity of 96.2% and specificity of 100%) [11]. By comparison, the optimal cut-off value for the Chinese CSI-9 was

18 points for the overall sample (sensitivity of 79.4% and specificity of 79.2%) and 19 points for the chronic pain group (sensitivity of 74.4% and specificity of 70.7%). Notably, the sensitivity, NPV, and accuracy of the Chinese CSI-9 were lower when a cut-off value of 20 points [11] was used, although the specificity and PPV were higher using a cut-off value of 20 points. The reasons underlying the difference in the optimal cut-off value between the Chinese and original versions of the CSI-9 remain undetermined, but the possibilities include differences in the characteristics of the study population (patients with chronic pain and healthy controls). Additional studies are required to confirm the optimal cut-off value for the Chinese CSI-9.

The management of chronic pain in China is still in its infancy [2, 5]. Indeed, although distressful and significantly affects the quality of life, chronic pain does not threaten the life of the patients, and healthcare priority is directed towards dangerous conditions. A major issue with chronic pain is that it is subjective, no device or blood test can quantify it, and its perception depends upon the individual. Such a situation calls for pragmatic evaluation tools for chronic pain, which are able to raise the diagnosis rate and further increase the treatment rate. The simplified CSI-9 that was validated in Chinese in this study is indeed convenient and easy to use in clinical practice. According to the results, CSI-9 was very suitable for the Chinese population. This validation and adaption of CSI-9 in China are of great significance for the improvement of future management strategies for chronic pain in China. The application of Chinese CSI-9 can help minimize the under-diagnosis rate, increase timely and appropriate treatments, alleviate patients' pain, and increase patient satisfaction.

This study has some limitations. This was a single-center study, so the results may not be generalized to other patients with chronic pain. The CSS-related disorders diagnosed in part B of the CSI-9 were self-reported and not confirmed by a clinician; moreover, some patients may have had other CSS-related conditions not included in the scale. The use of a self-reporting scale would be predicted to introduce response bias. Alternative methods (such as quantitative sensory testing) were not carried out to confirm the presence/absence of CS. The prefinal questionnaire was tested on only six participants instead of the 30–40 recommended by Beaton et al. [33] because of the limited patient population. The ability of the Chinese CSI-9 to detect CS was not assessed in a separate validation group. Some study patients were receiving therapy, and this may have decreased the severity of their chronic pain/CS symptoms. Finally, a responsiveness analysis was not carried out to evaluate the effects of treatment.

## Conclusion

In conclusion, a Chinese version of the CSI-9 was successfully generated using a process of translation, back-translation, and cultural adaptation. The Chinese CSI-9 had excellent test-retest reliability and satisfactory structural validity and construct validity. We propose that this simple scale could be used in China as a self-report questionnaire in clinical practice and research settings for screening CSS. Nevertheless, additional studies are required to confirm the optimal cut-off value of the CSI-9 before it can be used to screen for patients with CS.

## Supporting information

**S1 Data.**
(XLSX)

**S1 Table. Answers to part B of the Chinese 9-item central sensitization inventory.**
(DOCX)

**S2 Table. Answers to part A of the Chinese 9-item central sensitization inventory.**
(DOCX)

## Author Contributions

**Conceptualization:** Dongfeng Liang, Xiangli Yu, Xiaojie Guo, Ronghuan Jiang.

**Data curation:** Dongfeng Liang, Xiangli Yu, Xiaojie Guo, Jie Zhang.

**Formal analysis:** Dongfeng Liang, Xiangli Yu, Xiaojie Guo, Jie Zhang, Ronghuan Jiang.

**Investigation:** Dongfeng Liang, Xiangli Yu, Xiaojie Guo, Jie Zhang.

**Methodology:** Dongfeng Liang, Xiangli Yu, Xiaojie Guo.

**Project administration:** Dongfeng Liang, Ronghuan Jiang.

**Supervision:** Dongfeng Liang.

**Validation:** Dongfeng Liang.

**Writing – original draft:** Dongfeng Liang, Xiangli Yu, Xiaojie Guo.

**Writing – review & editing:** Dongfeng Liang, Xiangli Yu, Xiaojie Guo, Jie Zhang, Ronghuan Jiang.

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
