## [Decision Letter · Decision Letter 0]

4 Nov 2022

PONE-D-22-25986Cross-cultural adaptation and validation of the Chinese version of the short-form of the Central Sensitization Inventory (CSI-9) in patients with chronic pain: a single-center studyPLOS ONE

Dear Dr. Liang,

Thank you for submitting your manuscript to PLOS ONE. After careful consideration, we feel that it has merit but does not fully meet PLOS ONE’s publication criteria as it currently stands. Therefore, we invite you to submit a revised version of the manuscript that addresses the points raised during the review process.

Additional Editor Comments:

One of the reviewers recommended some important suggestions. Please carefully revise your manuscript. I will send the paper to the same reviewer.

We look forward to receiving your revised manuscript.

Kind regards,

Fatih Özden, PhD

Academic Editor

PLOS ONE

Journal Requirements:

Reviewers' comments:

Reviewer's Responses to Questions

**Comments to the Author**

1. Is the manuscript technically sound, and do the data support the conclusions?

Reviewer #1: Partly

Reviewer #2: Yes

2. Has the statistical analysis been performed appropriately and rigorously? 

Reviewer #1: No

Reviewer #2: Yes

3. Have the authors made all data underlying the findings in their manuscript fully available?

Reviewer #1: No

Reviewer #2: Yes

4. Is the manuscript presented in an intelligible fashion and written in standard English?

Reviewer #1: Yes

Reviewer #2: Yes

5. Review Comments to the Author

Reviewer #1: Thank you for inviting me to have the opportunity to review this manuscript. This is a validation study culturally adapting the English short form of the Central Sensitization Inventory (CSI) into simplified Chinese in patients with chronic pain. I have the following comments and suggestions for the authors:

Abstract

1. In the abstract, it was reported that “chronic pain affects more than 20% of the general population” while “up to 30% of people are affected by chronic pain” was stated in the Introduction. It would be better for the authors to unify the percentage of people affected by chronic pain in both the Abstract and Introduction.

Introduction

2. The authors should notice that there is already a culturally adapted version of Chinese Central Sensitization Inventory published recently (Feng et al, 2022).

3. Lines 61-64, the sentence is not clear and it is suggested to be re-written for more clarity, e.g. “The current situation regarding the diagnosis and management of chronic pain is not satisfactory, with under diagnosis and treatment. Patients with chronic pain may be neglected for pain management or provided with insufficient pain control.”

Methods

4. Lines 107-8, the details of recruitment of the healthy controls should be provided. What was the physical examination center? Was this center located inside the hospital? What were its functions …?

5. Lines 113-5, lumbago and back pain are literally synonymous; and elbow pain should have included lateral epicondylitis which is now more appropriately labelled as lateral epicondylalgia.

6. The specified inclusion criteria for the healthy controls did not exclude any acute pain status of the controls. So were there any healthy subjects with acute pain included in the study?

7. Lines 136-7 and 147-8, the citation of the “Brislin bidirectional translation method” should be provided.

8. As the forward and backward translations of the CSI-9 were fundamentally important to develop a culturally valid Chinese CSI-9, more details of the translation processes (forward and backward) such as the aim to identify any issues of ambiguous meanings in the original questionnaire or any inconsistencies or conceptual errors in the translations, and more details of the background of the translators in the forward and backward translations such as working experience as healthcare professionals, and qualifications of the professional translators … should be provided. Did the expert committee consider the semantic, idiomatic, experiential and conceptual equivalences as mentioned by Beaton et al (2000) (reference number 15) in finalizing the initial draft of Chinese CSI-9?

9. If the authors really followed the steps outlined by Beaton et al (2000), there should be 3 forward translations, T1, T2 and T3 and the 3 translators agreed a preliminary version A. Also, there should be 2 backward translations, BT1 and BT2. Was there a common backward translated English version BT1-2, or the expert committee reviewed the version A (common forward translation) with BT1 and BT2 to produce the pre-final version B (Chinese CSI-9)? Please clarify.

10. As recommended by Beaton et al (2000), the pre-final version should be administered to 30 to 40 subjects for pilot testing. The authors have to justify why only 6 patients were recruited to test the pre-final version for obtaining feedback or to address the limited number of subjects during pilot testing in the Limitations.

11. It is uncommon to use the Delphi method to develop the final version of questionnaire from the pre-final version with the feedback from the pilot tested subjects. The authors should provide more justification for the use of the Delphi method in this aspect.

12. Lines 168-70, what is the rationale or justification underlying the grading of subclinical (0-9 points), mild (10-19 points) and moderate/severe (>/=20 points) for the Chinese CSI-9 part A? Is there any reference for supporting this grading?

13. The citations and brief measurement properties of the Chinese Brief Pain Inventory and Chinese Pain Catastrophic Scale should be provided.

14. In the validity assessment, the authors had actually examined the construct validity of the Chinese CSI-9 in terms of structural/factorial validity, convergent validity and discriminant/divergent validity (Portney, 2020, p.127-140). It was mentioned that “The principal components were screened by promax rotation, and items were deleted if they had a factor loading <0.4” (lines 187-8). Was principal components analysis conducted in addition to the confirmatory factor analysis (CFA)? Why this had to be done if CFA was already planned?

15. In CFA, the number of factors and items loading on those factors had to be specified before conducting the CFA. What were the initial considerations in determining the number of factors and which items should be loaded to which factors in the CFA?

16. The criteria for interpreting those indices of model fit in CFA and magnitude of correlations in convergent validity should be specified in advance.

17. Line 212, the authors have to elaborate, how the “stable therapeutic regimen” could determine which patients were stable for the test-retest? Were these “stable patients” determined solely by the doctors or researchers, or by the self-report of the patients?

18. Lines 213-4, the “intra-group correlation coefficient” should be more correctly be “intraclass correlation coefficient” (ICC). The authors need to provide the choice of model for the ICC, e.g. one-way random-effects model, two-way random-effects model or two-way mixed effects model and whether “consistency” or “absolute agreement” had been used for the latter two models.

19. The authors should provide the sample size estimations for different parts of the cross-cultural adaptation of the CSI-9, e.g. how many subjects should be required for the confirmatory factor analysis, test-retest reliability analysis, correlational analyses and ROC curve analyses?

Results

20. It was mentioned that there were 290 patients recruited in the final analysis, with 235 patients with chronic pain and 55 healthy controls. Does this mean that those healthy controls were actually “patients”?

21. In Table 1, one decimal place should be good enough for presenting the scores of those variables to avoid false precision. The “overall population” should be “overall sample” or “all participants”.

22. In Table 2 showing the 9 English items of CSI-9, except the item “I do not sleep well”, all the other 8 items are slightly different from the original items of the CSI-25 published by Mayer et al (2012). Were those 8 English items modified during the background translation of the Chinese CSI-9?

23. How did those 3 factors determined with the loading of the 9 items? It seems inappropriate to label the Factor 3 as “headache/jaw symptoms” as there was no item related to jaw symptom in the Chinese CSI-9.

24. The author should report the model fit indices of the CFA, e.g. GFI, AGFI, CFI, TLI and RMSEA in the Results.

25. In Table 3, did the result of correlations meet the pre-specified criteria for convergent and divergent validities of the Chinese CSI-9 with those variables?

26. The model of ICC should be specified in reporting the ICC values.

27. Cronbach’s alpha is a measure of the average inter-relatedness of items of a scale examined for the internal consistency of the scale. In Table 6, it appears that each of the 9 items had their own Cronbach’s alpha which is unconceivable as single item would not have correlation with other items!

Discussion

28. Lines 355-6, it is more appropriate to state that “structural validity” refers to the extent to which the scores of a scale are an adequate reflection of the dimensionality of the construct to be measured (Mokkink et al, 2010).

29. The development of the CSI-9 by Nishigami et al (2018) was resulted from the use of Rasch analysis to achieve unidimensionality, i.e. single factor. The authors should discuss whether they had conducted the CFA with single factor in mind and how they would end up with 3 factors from the CFA, contrary to the result of Nishigami et al’s study (2018).

30. The items of the original CSI-25 were developed by an interdisciplinary team of healthcare professionals but there was no description of how those items were identified or developed from any item pool extracted from the literature. There were no inputs and cognitive debriefing from the patients with chronic pain and central sensitization during the development phase of the original CSI-25. Therefore, the content validity of the CSI-25 may have limitations which may partly account for the different factor models found in different populations. The development of a short form of an original questionnaire would require rigorous methodology; otherwise the validity of the original questionnaire (especially content validity) may further be compromised (Goetz et al, 2013)! The authors should have a good discussion on these issues.

31. Lines 433-4, the application of Chinese CSI-9 should not be intended to “increase diagnosis rate” but to minimize “under-diagnosis” of central sensitization so that patients with central sensitization can receive timely and appropriate treatment.

32. Lines 436-7, the results may not “be generalized” to other patients with chronic pain.

Conclusion

33. Lines 452-3, it is suggested to rephrase the sentence as, “We propose that this simple scale could be used in China as a self-report questionnaire in clinical practice and research settings for screening central sensitization syndrome”.

References

Beaton DE, Bombardier C, Guillemin F, Ferraz MB. Guidelines for the process of cross-cultural adaptation of self-report measures. Spine 2000; 25(24): 3186-91. doi: 10.1097/00007632-200012150-00014.

Feng B, Hu X, Lu WW, Wang Y, Ip WY. Cultural Validation of the Chinese Central Sensitization Inventory in patients with chronic pain and its predictive ability of comorbid central sensitivity syndromes. J Pain Res 2022; 15: 467-477. doi: 10.2147/JPR.S348842.

Goetz C, Coste J, Lemetayer F, Rat AC, Montel S, Recchia S, Debouverie M, Pouchot J, Spitz E, Guillemin F. Item reduction based on rigorous methodological guidelines is necessary to maintain validity when shortening composite measurement scales. J Clin Epidemiol 2013; 66(7): 710-8. doi: 10.1016/j.jclinepi.2012.12.015.

Mayer TG, Neblett R, Cohen H, Howard KJ, Choi YH, Williams MJ, Perez Y, Gatchel RJ. The development and psychometric validation of the central sensitization inventory. Pain Pract 2012; 12(4): 276-85. doi: 10.1111/j.1533-2500.2011.00493.x.

Mokkink LB, Terwee CB, Patrick DL, Alonso J, Stratford PW, Knol DL, Bouter LM, de Vet HC. The COSMIN study reached international consensus on taxonomy, terminology, and definitions of measurement properties for health-related patient-reported outcomes. J Clin Epidemiol 2010; 63(7): 737-45. doi: 10.1016/j.jclinepi.2010.02.006.

Nishigami T, Tanaka K, Mibu A, Manfuku M, Yono S, Tanabe A. Development and psychometric properties of short form of central sensitization inventory in participants with musculoskeletal pain: A cross-sectional study. PLoS One 2018;13(7): e0200152. doi: 10.1371/journal.pone.0200152.

Portney LG. Foundations of Clinical Research: Applications to Evidence-Based Practice, 4th ed, Philadelphia: F.A. Davis, 2020.

Reviewer #2: All in all, a very well written validity and reliability article. Just ı can say more healthy controls would be better. Congratulations to the authors.

6. PLOS authors have the option to publish the peer review history of their article (what does this mean?). If published, this will include your full peer review and any attached files.

Reviewer #1: **Yes: **Raymond CC Tsang

Reviewer #2: **Yes: **Esedullah AKARAS

---

## [Author Response · Author response to Decision Letter 0]

6 Dec 2022

Manuscript ID: PONE-D-22-25986

Title: Cross-cultural adaptation and validation of the Chinese version of the short-form of the Central Sensitization Inventory (CSI-9) in patients with chronic pain: a single-center study

Name of Journal: Plos One

Response to Reviewers’ comments

Dear Editor PhD. Fatih Özden:

We thank you for carefully considering our manuscript, and we are grateful for your response and overall positive feedback. After carefully reviewing the comments made by the reviewers, we have modified the manuscript to improve the presentation and discussion of our study, providing a full context to the study. 

Furthermore, we have thoroughly revised the manuscript and improved the language quality, taking into consideration the reviewers’ and editors’ comments. We hope that you now find the revised manuscript suitable for publication and look forward to contributing to your journal. We believe that the revised manuscript would be of interest to your readership.

Please do not hesitate to contact me with any questions or concerns regarding the current manuscript.

Best regards,

Dong-feng Liang

 

Reviewer #1

Thank you for inviting me to have the opportunity to review this manuscript. This is a validation study culturally adapting the English short form of the Central Sensitization Inventory (CSI) into simplified Chinese in patients with chronic pain. I have the following comments and suggestions for the authors: 

Abstract

1. In the abstract, it was reported that “chronic pain affects more than 20% of the general population” while “up to 30% of people are affected by chronic pain” was stated in the Introduction. It would be better for the authors to unify the percentage of people affected by chronic pain in both the Abstract and Introduction.

Response: We thank the Reviewer for pointing out that discrepancy. It was corrected.

Introduction

2. The authors should notice that there is already a culturally adapted version of Chinese Central Sensitization Inventory published recently (Feng et al, 2022).

Response: We thank the Reviewer for the comment. China is a huge country with many research teams in various regions. We were not aware that someone else was working on a Chinese CSI-9 version, and their paper was published after we finalized the present manuscript. Of note, they validated a Chinese version of the original CSI-25 [1], while we directly validated the CSI-9 in Chinese. Nevertheless, we added that paper to our manuscript.

3. Lines 61-64, the sentence is not clear and it is suggested to be re-written for more clarity, e.g. “The current situation regarding the diagnosis and management of chronic pain is not satisfactory, with under diagnosis and treatment. Patients with chronic pain may be neglected for pain management or provided with insufficient pain control.”

Response: We thank the Reviewer. We revised that part of the Introduction.

Original version: In China, since the condition does not threaten the life of the patients, chronic pain is often overlooked. The current situation regarding the diagnosis and treatment of chronic pain is not satisfying, with low diagnosis rate, low treatment rate, insufficient pain control, poor management of pain, and neglect of pain management.

Revised version: In China, since chronic pain does not threaten the life of the patients, it is often overlooked, leading to low diagnosis rates, low treatment rates, insufficient pain control, poor management of pain, and neglect of pain management.

Methods

4. Lines 107-8, the details of recruitment of the healthy controls should be provided. What was the physical examination center? Was this center located inside the hospital? What were its functions …?

Response: The physical examination center is the department located inside the hospital that specializes in physical examination, and it offers routine regular physical examinations for ordinary people. It was clarified in the Methods-study participants section.

5. Lines 113-5, lumbago and back pain are literally synonymous; and elbow pain should have included lateral epicondylitis which is now more appropriately labelled as lateral epicondylalgia.

Response: The Reviewer is right. These were corrected as suggested.

6. The specified inclusion criteria for the healthy controls did not exclude any acute pain status of the controls. So were there any healthy subjects with acute pain included in the study?

Response: CSI focuses on chronic pain, not related to acute pain. Neither the chronic population nor healthy population in our study excluded acute pain. Whether there was acute pain did not affect the validity test or reliability test of CSI. The BPI of four individuals in the healthy population showed there had acute pain recently.

7. Lines 136-7 and 147-8, the citation of the “Brislin bidirectional translation method” should be provided.

Response: Brislin, R. W. (1970). Back-Translation for Cross-Cultural Research. Journal of Cross-Cultural Psychology, 1(3), 185–216. https://doi.org/10.1177/135910457000100301. It was added to the manuscript.

8. As the forward and backward translations of the CSI-9 were fundamentally important to develop a culturally valid Chinese CSI-9, more details of the translation processes (forward and backward) such as the aim to identify any issues of ambiguous meanings in the original questionnaire or any inconsistencies or conceptual errors in the translations, and more details of the background of the translators in the forward and backward translations such as working experience as healthcare professionals, and qualifications of the professional translators … should be provided. Did the expert committee consider the semantic, idiomatic, experiential and conceptual equivalences as mentioned by Beaton et al (2000) (reference number 15) in finalizing the initial draft of Chinese CSI-9?

Response: The experts were all experienced in fibromyalgia, pain, and the psychology of pain:

1). Dongfeng Liang, Deputy Chief Physician, Department of Rheumatology and Immunology, First Medical Center, PLA General Hospital;

2). Wu Qingjun, Chief Physician of the Rheumatology and Immunology Department of Peking Union Medical College Hospital;

3). Luo Fang, Chief Physician of the Pain Department of Tiantan Hospital Affiliated to Capital Medical University;

4). Shi Hui, Deputy Chief Physician of the Psychiatry Department of Chaoyang Hospital Affiliated to Capital Medical University;

5). Yao Zhongqiang, Deputy Chief Physician of the Rheumatology and Immunology Department of the Third Affiliated Hospital of Beijing Medical University;

6). Xu Xiaoyan, Deputy Chief Physician of the Rheumatology and Immunology Department of Zhongda Hospital Affiliated to Southeast University;

7). Li Ling, Deputy Chief Physician of the Rheumatology and Immunization Department of Guangdong Provincial People’s Hospital.

All the above experts participated in the Delphi consultation on the Chinese version. With regard to the translation process, the implementation process was as follows. 

1) Two experts (LD and ZG) and a Chinese professional translator (Shanghai Richard Translation Co.) translated the scale from English to Chinese, and the researchers integrated it into the original Chinese version. 

2) A professional translator of Chinese nationality and a professional English native-speaking translator (Shanghai Richard Translation Co.) back-translated the original Chinese version of the scale, namely, Chinese to English. 

3) The researchers combined the results of the English-to-Chinese translation and Chinese-to-English translation to form the first Chinese version of the scale again. 

4) The first Chinese version of the scale was tested by doctors and patients, and the translated items were modified again according to the test results to obtain the final version of the Chinese CSI-9.

9. If the authors really followed the steps outlined by Beaton et al (2000), there should be 3 forward translations, T1, T2 and T3 and the 3 translators agreed a preliminary version A. Also, there should be 2 backward translations, BT1 and BT2. Was there a common backward translated English version BT1-2, or the expert committee reviewed the version A (common forward translation) with BT1 and BT2 to produce the pre-final version B (Chinese CSI-9)? Please clarify.

Response: We used the Brislin bidirectional translation method [2]. All the above experts participated in the Delphi consultation on the Chinese version. The implementation of the translation process was described as in the above response.

10. As recommended by Beaton et al (2000), the pre-final version should be administered to 30 to 40 subjects for pilot testing. The authors have to justify why only 6 patients were recruited to test the pre-final version for obtaining feedback or to address the limited number of subjects during pilot testing in the Limitations.

Response: We thank the Reviewer for the comment. A pre-study of 30-40 was impossible in our clinical setting, and the time required to perform it would have incurred the risk of making the first participants obsolete by the time all participants completed the questionnaires. Therefore, we administered the questionnaire to 10 participants. It was added as a limitation.

11. It is uncommon to use the Delphi method to develop the final version of questionnaire from the pre-final version with the feedback from the pilot tested subjects. The authors should provide more justification for the use of the Delphi method in this aspect.

Response: The Delphi method was applied in order to consult as many experienced experts in various hospitals across China as possible (seven experts in total), thus increasing the usability and readability of Chinese CSI-9. Furthermore, the results from six patients were also considered. All results from the doctors and patients were fully discussed to achieve the final Chinese CSI-9.

12. Lines 168-70, what is the rationale or justification underlying the grading of subclinical (0-9 points), mild (10-19 points) and moderate/severe (>/=20 points) for the Chinese CSI-9 part A? Is there any reference for supporting this grading?

Response: The cutoff points were according to the original CSI-9 version [3]. We added the reference to the manuscript.

13. The citations and brief measurement properties of the Chinese Brief Pain Inventory and Chinese Pain Catastrophic Scale should be provided.

Response: The citations and measurements of Chinese BPI and Chinese PCS scales were clarified in the “Data collection” section and the “Validity assessment” section in Methods. 

Wang XS, Mendoza TR, Gao SZ, Cleeland CS. The Chinese version of the Brief Pain Inventory (BPI-C): its development and use in a study of cancer pain. Pain. 1996 Oct;67(2-3):407-16. doi: 10.1016/0304-3959(96)03147-8. PMID: 8951936.

Shen B, Wu B, Abdullah TB, Zhan G, Lian Q, Vania Apkarian A, Huang L. Translation and validation of Simplified Chinese version of the Pain Catastrophizing Scale in chronic pain patients: Education may matter. Mol Pain. 2018 Jan-Dec;14:1744806918755283. doi: 10.1177/1744806918755283. Epub 2018 Jan 21. PMID: 29353539; PMCID: PMC5788090. 

14. In the validity assessment, the authors had actually examined the construct validity of the Chinese CSI-9 in terms of structural/factorial validity, convergent validity and discriminant/divergent validity (Portney, 2020, p.127-140). It was mentioned that “The principal components were screened by promax rotation, and items were deleted if they had a factor loading <0.4” (lines 187-8). Was principal components analysis conducted in addition to the confirmatory factor analysis (CFA)? Why this had to be done if CFA was already planned?

Response: The CFA results indicated no need to delete items. EFA delete items by principal components analysis. If the result of CFA was not satisfying, then EFA was conducted. Principal components analysis was not conducted in this study. The mistake occurred during manuscript preparation and translation. It has been deleted to avoid misunderstanding. 

15. In CFA, the number of factors and items loading on those factors had to be specified before conducting the CFA. What were the initial considerations in determining the number of factors and which items should be loaded to which factors in the CFA?

Response: The Chinese CSI-9 was a validation of the Chinese version from the original English CSI-9. All items and factors were determined according to the original version. 

16. The criteria for interpreting those indices of model fit in CFA and magnitude of correlations in convergent validity should be specified in advance.

Response: The criteria for interpreting model fit and magnitude of correlations in convergent validity were specified in advance. It was clarified in the Methods section.

17. Line 212, the authors have to elaborate, how the “stable therapeutic regimen” could determine which patients were stable for the test-retest? Were these “stable patients” determined solely by the doctors or researchers, or by the self-report of the patients?

Response: Test-retest reliability was used to assess the stability of the CSI-9. When tested for the first time, all subjects were confirmed by randomization whether to be tested for the second time at 7 ± 1 days after the first test. Those who are retested were required to maintain a stable therapeutic regimen (i.e., no changes in drugs and dosages) between the first and second tests. It was clarified in the “Reliability assessment” section.

18. Lines 213-4, the “intra-group correlation coefficient” should be more correctly be “intraclass correlation coefficient” (ICC). The authors need to provide the choice of model for the ICC, e.g. one-way random-effects model, two-way random-effects model or two-way mixed effects model and whether “consistency” or “absolute agreement” had been used for the latter two models.

Response: It was corrected into “intraclass correlation coefficient (ICC)”. We used “two-way random-effects model” and “absolute agreement”. It was clarified in the manuscript.

19. The authors should provide the sample size estimations for different parts of the cross-cultural adaptation of the CSI-9, e.g. how many subjects should be required for the confirmatory factor analysis, test-retest reliability analysis, correlational analyses and ROC curve analyses?

Response: The sample size calculation was based on CFA, as designed in the protocol, and including at least 200 patients would meet the needs for validation, and 50 would be sufficient for retest reliability. The necessary sample sizes for other measurements were all not that large as CFA. 

Results

20. It was mentioned that there were 290 patients recruited in the final analysis, with 235 patients with chronic pain and 55 healthy controls. Does this mean that those healthy controls were actually “patients”?

Response: We apologize for the wrong choice of word. In fact, the healthy controls were included from the physical examination department who were intended for a routine health check. It was corrected.

21. In Table 1, one decimal place should be good enough for presenting the scores of those variables to avoid false precision. The “overall population” should be “overall sample” or “all participants”.

Response: We thank the Reviewer. The tables were revised accordingly.

22. In Table 2 showing the 9 English items of CSI-9, except the item “I do not sleep well”, all the other 8 items are slightly different from the original items of the CSI-25 published by Mayer et al (2012). Were those 8 English items modified during the background translation of the Chinese CSI-9?

Response: Indeed, the items were modified during the translation process. To avoid misunderstanding, we revised the items in Tables 2, 4, and 6, and kept them the same with those in Mayer et al (2012).

23. How did those 3 factors determined with the loading of the 9 items? It seems inappropriate to label the Factor 3 as “headache/jaw symptoms” as there was no item related to jaw symptom in the Chinese CSI-9.

Response: We were wrong to label the three factors. According to the original CSI-9, we redid the statistical analysis as one factor as follows:

Structural validity

 CSI items Mean (SD) Factor 1

1 Unrefreshed in morning 2.2±1.3 0.692

2 Muscles stiff/achy 2.5±1.4 0.773

3 Pain all over body 2.2±1.5 0.865

4 Headaches 1.3±1.2 0.567

5 Do not sleep well 2.1±1.4 0.711

6 Difficulty concentrating 1.6±1.3 0.712

7 Stress makes symptoms worse 1.5±1.3 0.692

8 Tension neck and shoulder 2.3±1.4 0.752

9 Poor memory 2.1±1.3 0.623

24. The author should report the model fit indices of the CFA, e.g. GFI, AGFI, CFI, TLI and RMSEA in the Results.

Response: We thank the Reviewer. The model fit indices were chi-square (CMIN)=60.05, degree of freedom (DF)=24, P<0.001, CMIN/DF=2.877, Tucker-Lewis index (TLI)=0.948, comparative fit index (CFI)=0.965, root mean square error of approximation (RMSEA)=0.081. They were added to the results.

25. In Table 3, did the result of correlations meet the pre-specified criteria for convergent and divergent validities of the Chinese CSI-9 with those variables?

Response: We thank the Reviewer. Yes, the result of the correlations met the pre-specified criteria for convergent and divergent validities of the Chinese CSI-9 with those variables.

26. The model of ICC should be specified in reporting the ICC values.

Response: We thank the Reviewer. It was the “two-way random-effects model”. It was clarified in the “Test-retest reliability” in the Results.

27. Cronbach’s alpha is a measure of the average inter-relatedness of items of a scale examined for the internal consistency of the scale. In Table 6, it appears that each of the 9 items had their own Cronbach’s alpha which is unconceivable as single item would not have correlation with other items!

Response: The Cronbach’s alpha displayed in Table 6 was not for each item itself, and it was the total Cronbach’s alpha of CSI-9 except the corresponding item. It was clarified in the text. The Greek adaption of CSI-25 also displayed like this. 

Bilika P, Neblett R, Georgoudis G, Dimitriadis Z, Fandridis E, Strimpakos N, Kapreli E. Cross-cultural Adaptation and Psychometric Properties of the Greek Version of the Central Sensitization Inventory. Pain Pract. 2020 Feb;20(2):188-196. doi: 10.1111/papr.12843. Epub 2020 Jan 6. PMID: 31605651.

Discussion

28. Lines 355-6, it is more appropriate to state that “structural validity” refers to the extent to which the scores of a scale are an adequate reflection of the dimensionality of the construct to be measured (Mokkink et al, 2010).

Response: We thank the Reviewer. It was revised as suggested.

29. The development of the CSI-9 by Nishigami et al (2018) was resulted from the use of Rasch analysis to achieve unidimensionality, i.e. single factor. The authors should discuss whether they had conducted the CFA with single factor in mind and how they would end up with 3 factors from the CFA, contrary to the result of Nishigami et al’s study (2018).

Response: We thank the Reviewer. We were wrong. It was to be a single factor. We redid the calculation and provided the results above. It was corrected.

30. The items of the original CSI-25 were developed by an interdisciplinary team of healthcare professionals but there was no description of how those items were identified or developed from any item pool extracted from the literature. There were no inputs and cognitive debriefing from the patients with chronic pain and central sensitization during the development phase of the original CSI-25. Therefore, the content validity of the CSI-25 may have limitations which may partly account for the different factor models found in different populations. The development of a short form of an original questionnaire would require rigorous methodology; otherwise the validity of the original questionnaire (especially content validity) may further be compromised (Goetz et al, 2013)! The authors should have a good discussion on these issues.

Response: We thank the Reviewer, but the Chinese CSI-9 was not a simplified version of the Chinese CSI-25. It is a translation of the validated original CSI-9.

31. Lines 433-4, the application of Chinese CSI-9 should not be intended to “increase diagnosis rate” but to minimize “under-diagnosis” of central sensitization so that patients with central sensitization can receive timely and appropriate treatment.

Response: We agree with the Reviewer. It was revised as suggested.

32. Lines 436-7, the results may not “be generalized” to other patients with chronic pain.

 Response: It was corrected.

Conclusion

33. Lines 452-3, it is suggested to rephrase the sentence as, “We propose that this simple scale could be used in China as a self-report questionnaire in clinical practice and research settings for screening central sensitization syndrome”.

Response: We thank the Reviewer. It was revised as suggested.

References

Beaton DE, Bombardier C, Guillemin F, Ferraz MB. Guidelines for the process of cross-cultural adaptation of self-report measures. Spine 2000; 25(24): 3186-91. doi: 10.1097/00007632-200012150-00014.

Feng B, Hu X, Lu WW, Wang Y, Ip WY. Cultural Validation of the Chinese Central Sensitization Inventory in patients with chronic pain and its predictive ability of comorbid central sensitivity syndromes. J Pain Res 2022; 15: 467-477. doi: 10.2147/JPR.S348842.

Goetz C, Coste J, Lemetayer F, Rat AC, Montel S, Recchia S, Debouverie M, Pouchot J, Spitz E, Guillemin F. Item reduction based on rigorous methodological guidelines is necessary to maintain validity when shortening composite measurement scales. J Clin Epidemiol 2013; 66(7): 710-8. doi: 10.1016/j.jclinepi.2012.12.015.

Mayer TG, Neblett R, Cohen H, Howard KJ, Choi YH, Williams MJ, Perez Y, Gatchel RJ. The development and psychometric validation of the central sensitization inventory. Pain Pract 2012; 12(4): 276-85. doi: 10.1111/j.1533-2500.2011.00493.x.

Mokkink LB, Terwee CB, Patrick DL, Alonso J, Stratford PW, Knol DL, Bouter LM, de Vet HC. The COSMIN study reached international consensus on taxonomy, terminology, and definitions of measurement properties for health-related patient-reported outcomes. J Clin Epidemiol 2010; 63(7): 737-45. doi: 10.1016/j.jclinepi.2010.02.006.

Nishigami T, Tanaka K, Mibu A, Manfuku M, Yono S, Tanabe A. Development and psychometric properties of short form of central sensitization inventory in participants with musculoskeletal pain: A cross-sectional study. PLoS One 2018;13(7): e0200152. doi: 10.1371/journal.pone.0200152.

Portney LG. Foundations of Clinical Research: Applications to Evidence-Based Practice, 4th ed, Philadelphia: F.A. Davis, 2020.

Response: We thank the Reviewer. We carefully read the above references and we added some of them into the manuscript where appropriate.

Reviewer #2: 

All in all, a very well written validity and reliability article. Just ı can say more healthy controls would be better. Congratulations to the authors.

Response: We thank the Reviewer for the comment. We revised the whole manuscript according to your Reviewers’ comments.

References

1. Feng B, Hu X, Lu WW, Wang Y, Ip WY. Cultural Validation of the Chinese Central Sensitization Inventory in Patients with Chronic Pain and its Predictive Ability of Comorbid Central Sensitivity Syndromes. J Pain Res. 2022;15:467-77. Epub 2022/02/26. doi: 10.2147/JPR.S348842. PubMed PMID: 35210847; PubMed Central PMCID: PMCPMC8857991.

2. Brislin RW. Back-Translation for Cross-Cultural Research. J Cross Cultural Psychol. 1970;1(3):185-216. doi: 10.1177/135910457000100301.

3. Nishigami T, Tanaka K, Mibu A, Manfuku M, Yono S, Tanabe A. Development and psychometric properties of short form of central sensitization inventory in participants with musculoskeletal pain: A cross-sectional study. PLoS One. 2018;13(7):e0200152. Epub 2018/07/06. doi: 10.1371/journal.pone.0200152. PubMed PMID: 29975754; PubMed Central PMCID: PMCPMC6033441.

---

## [Decision Letter · Decision Letter 1]

12 Dec 2022

PONE-D-22-25986R1Cross-cultural adaptation and validation of the Chinese version of the short-form of the Central Sensitization Inventory (CSI-9) in patients with chronic pain: a single-center studyPLOS ONE

Dear Dr. Liang,

Thank you for submitting your manuscript to PLOS ONE. After careful consideration, we feel that it has merit but does not fully meet PLOS ONE’s publication criteria as it currently stands. Therefore, we invite you to submit a revised version of the manuscript that addresses the points raised during the review process.

Additional Editor Comments:

The reviewer suggested a minor revision. I look forward to your revised article.

We look forward to receiving your revised manuscript.

Kind regards,

Fatih Özden, PhD

Academic Editor

PLOS ONE

Journal Requirements:

Reviewers' comments:

Reviewer's Responses to Questions

**Comments to the Author**

1. If the authors have adequately addressed your comments raised in a previous round of review and you feel that this manuscript is now acceptable for publication, you may indicate that here to bypass the “Comments to the Author” section, enter your conflict of interest statement in the “Confidential to Editor” section, and submit your "Accept" recommendation.

Reviewer #1: (No Response)

2. Is the manuscript technically sound, and do the data support the conclusions?

Reviewer #1: Yes

3. Has the statistical analysis been performed appropriately and rigorously? 

Reviewer #1: Yes

4. Have the authors made all data underlying the findings in their manuscript fully available?

Reviewer #1: No

5. Is the manuscript presented in an intelligible fashion and written in standard English?

Reviewer #1: Yes

6. Review Comments to the Author

Reviewer #1: Most of the issues have been addressed by the authors, except the following:

1. In the Abstract (lines 33-35) and Discussion (lines 366-367), it is still stated that “confirmatory factor analysis extracted three main factors (“physical symptoms”, “emotional distress”, and “headache/jaw symptoms”)”. Please clarify and correct this.

2. In Table 4, test-retest reliability, were the ICC values shown for each item resulted from the remaining items after deletion of that item? If that is the case, this should be explained in the footnote of the table for clarity. The same explanation should also be applied in Table 6 for the internal consistency.

3. The number of participants for pilot testing of the Chinese CSI-9 is 10 in the Discussion section (line 453) while this is reported to be 6 in the Methods section (line 164). Please clarify and correct this.

7. PLOS authors have the option to publish the peer review history of their article (what does this mean?). If published, this will include your full peer review and any attached files.

Reviewer #1: **Yes: **TSANG Chi-Chung Raymond

---

## [Author Response · Author response to Decision Letter 1]

3 Jan 2023

Manuscript ID: PONE-D-22-25986

Title: Cross-cultural adaptation and validation of the Chinese version of the short-form of the Central Sensitization Inventory (CSI-9) in patients with chronic pain: a single-center study

Name of Journal: PLoS One

Response to Reviewers’ comments

Dear Editor PhD. Fatih Özden:

Thank you for carefully considering our manuscript! We are grateful for your response and overall positive feedback. After carefully reviewing the comments made by Reviewer 1, we have modified the manuscript, providing a full context to the study. 

We hope that you now find the revised manuscript suitable for publication and look forward to contributing to your journal. We believe that the revised manuscript would be of interest to your readership.

Please do not hesitate to contact me in case of any questions or concerns regarding the current manuscript.

Best regards,

Dong-feng Liang

 

Reviewer #1

Reviewer #1: Most of the issues have been addressed by the authors, except the following:

1. In the Abstract (lines 33-35) and Discussion (lines 366-367), it is still stated that “confirmatory factor analysis extracted three main factors (“physical symptoms”, “emotional distress”, and “headache/jaw symptoms”)”. Please clarify and correct this. Response: We thank the Reviewer for pointing out that discrepancy. It was corrected.

2. In Table 4, test-retest reliability, were the ICC values shown for each item resulted from the remaining items after deletion of that item? If that is the case, this should be explained in the footnote of the table for clarity. The same explanation should also be applied in Table 6 for the internal consistency.

Response: The Reviewer is right. It was added as a footnote to Tables 4 and 6.

3. The number of participants for pilot testing of the Chinese CSI-9 is 10 in the Discussion section (line 453) while this is reported to be 6 in the Methods section (line 164). Please clarify and correct this.

Response: We thank the Reviewer for pointing out that discrepancy. It was corrected in the Discussion. It was six.

---

## [Decision Letter · Decision Letter 2]

9 Jan 2023

PONE-D-22-25986R2Cross-cultural adaptation and validation of the Chinese version of the short-form of the Central Sensitization Inventory (CSI-9) in patients with chronic pain: a single-center studyPLOS ONE

Dear Dr. Liang,

Thank you for submitting your manuscript to PLOS ONE. After careful consideration, we feel that it has merit but does not fully meet PLOS ONE’s publication criteria as it currently stands. Therefore, we invite you to submit a revised version of the manuscript that addresses the points raised during the review process.

We look forward to receiving your revised manuscript.

Kind regards,

Fatih Özden, PhD

Academic Editor

PLOS ONE

Journal Requirements:

Reviewers' comments:

Reviewer's Responses to Questions

**Comments to the Author**

1. If the authors have adequately addressed your comments raised in a previous round of review and you feel that this manuscript is now acceptable for publication, you may indicate that here to bypass the “Comments to the Author” section, enter your conflict of interest statement in the “Confidential to Editor” section, and submit your "Accept" recommendation.

Reviewer #1: (No Response)

2. Is the manuscript technically sound, and do the data support the conclusions?

Reviewer #1: Yes

3. Has the statistical analysis been performed appropriately and rigorously? 

Reviewer #1: Yes

4. Have the authors made all data underlying the findings in their manuscript fully available?

Reviewer #1: No

5. Is the manuscript presented in an intelligible fashion and written in standard English?

Reviewer #1: Yes

6. Review Comments to the Author

Reviewer #1: All the concerns have been addressed except the criteria of model fit indices for the cofirmatory factor analysis (CFA) and their citation are missing in the Methods - Validity Assessment section. The authors should provide this information to allow readers to evaluate the model fit of the one-factor model from the CFA.

7. PLOS authors have the option to publish the peer review history of their article (what does this mean?). If published, this will include your full peer review and any attached files.

Reviewer #1: **Yes: **Raymond CC Tsang

---

## [Author Response · Author response to Decision Letter 2]

13 Feb 2023

Manuscript ID: PONE-D-22-25986R2

Title: Cross-cultural adaptation and validation of the Chinese version of the short-form of the Central Sensitization Inventory (CSI-9) in patients with chronic pain: a single-center study

Name of Journal: PLoS One

Response to Reviewers’ comments

Dear Editor PhD. Fatih Özden:

Thank you for carefully considering our manuscript! We are grateful for your response and overall positive feedback. After carefully reviewing the comments made by Reviewer 1, we have modified the manuscript, providing a full context to the study. 

We hope that you now find the revised manuscript suitable for publication and look forward to contributing to your journal. We believe that the revised manuscript would be of interest to your readership.

Please do not hesitate to contact me in case of any questions or concerns regarding the current manuscript.

Best regards,

Dong-feng Liang

 

Reviewer #1

All the concerns have been addressed except the criteria of model fit indices for the confirmatory factor analysis (CFA) and their citation are missing in the Methods - Validity Assessment section. The authors should provide this information to allow readers to evaluate the model fit of the one-factor model from the CFA. Response: We thank the Reviewer. It was added in the Methods section.

Revised version: The indices used to determine the model fit were: chi-square degrees of freedom (�2/df), goodness-of-fit index (GFI), adjusted goodness-of-fit index (AGFI), comparative fit index (CFI), Tucker-Lewis Coefficient (TLI) and root mean square error of approximation (RMSEA). A model with �2/df<3, RMSEA<0.08, GFI>0.90, CFI>0.90, and TLI>0.90 suggested a good fit [reference].

Reference: Qiu H, Huang C, Liu Q, Jiang L, Xue Y, Wu W, Huang Z, Xu J. Reliability and validity of the Healthy Fitness Measurement Scale Version 1.0 (HFMS V1.0) in Chinese people. BMJ Open. 2021 Dec 7;11(12):e048269.

---

## [Editor Report · Decision Letter 3]

15 Feb 2023

Cross-cultural adaptation and validation of the Chinese version of the short-form of the Central Sensitization Inventory (CSI-9) in patients with chronic pain: a single-center study

PONE-D-22-25986R3

Dear Dr. Liang,

We’re pleased to inform you that your manuscript has been judged scientifically suitable for publication and will be formally accepted for publication once it meets all outstanding technical requirements.

Kind regards,

Fatih Özden, PhD

Academic Editor

PLOS ONE
---

## [Editor Report · Acceptance letter]

7 Mar 2023

PONE-D-22-25986R3 

Cross-cultural adaptation and validation of the Chinese version of the short-form of the Central Sensitization Inventory (CSI-9) in patients with chronic pain: a single-center study 

Dear Dr. Liang:

I'm pleased to inform you that your manuscript has been deemed suitable for publication in PLOS ONE. Congratulations! Your manuscript is now with our production department. 

Kind regards, 

on behalf of

Dr. Fatih Özden 

Academic Editor

PLOS ONE